# Mapping nanocrystalline disorder within an amorphous metal–organic framework

Adam F. Sapnik [1], Chao Sun[2], Joonatan E. M. Laulainen[1], Duncan N. Johnstone[1], Rik Brydson[2], Timothy Johnson [3], Paul A. Midgley[1], Thomas D. Bennett [1] & Sean M. Collins [1,2,4✉]

Intentionally disordered metal–organic frameworks (MOFs) display rich functional behaviour. However, the characterisation of their atomic structures remains incredibly challenging. X-ray pair distribution function techniques have been pivotal in determining their average local structure but are largely insensitive to spatial variations in the structure. Fe-BTC (BTC = 1,3,5-benzenetricarboxylate) is a nanocomposite MOF, known for its catalytic properties, comprising crystalline nanoparticles and an amorphous matrix. Here, we use scanning electron diffraction to first map the crystalline and amorphous components to evaluate domain size and then to carry out electron pair distribution function analysis to probe the spatially separated atomic structure of the amorphous matrix. Further Bragg scattering analysis reveals systematic orientational disorder within Fe-BTC's nanocrystallites, showing over 10° of continuous lattice rotation across single particles. Finally, we identify candidate unit cells for the crystalline component. These independent structural analyses quantify disorder in Fe-BTC at the critical length scale for engineering composite MOF materials.

[1] Department of Materials Science and Metallurgy, University of Cambridge, Cambridge, UK. [2] School of Chemical and Process Engineering, University of Leeds, Leeds, UK. [3] Johnson Matthey Technology Centre, Blount's Court, Sonning Common, Reading, UK. [4] School of Chemistry, University of Leeds, Leeds, UK. ✉email: S.M.Collins@leeds.ac.uk

Amorphous metal–organic frameworks (MOFs) are identified by the absence of Bragg scattering in their diffraction patterns[1]. However, these diffraction patterns tell us very little about the actual atomic structure. An amorphous MOF may be topologically disordered, possessing a structure that is inherently aperiodic, as is the case for $a$ZIF-4[2]. However, we cannot assume all amorphous MOFs are topologically disordered; nanostructured materials can also possess very broad diffraction patterns that may also appear to be amorphous[3,4]. In addition, MOFs may exhibit all manner of other types of disorder—such as defects (e.g., missing linkers or missing nodes), static, dynamic and low-dimensional disorder[5,6]. Simply denoting these materials as "amorphous" overlooks the structural complexity that they may contain. In order to gain a true atomistic understanding of amorphous MOFs, we must characterise their structures over multiple length scales, determining the nature and extent to which they are ordered/disordered.

The structural complexity of amorphous MOFs often affords highly functional materials[1,7,8]. Amorphous MOFs have found applications in the irreversible trapping of toxic guest species and in the potential delivery of anti-cancer drug molecules with tuneable release profiles[9,10]. In addition, they have been reported to possess powerful catalytic sites, isotropic conducting paths, and enhanced ion dynamics over their crystalline counterparts[8]. Understanding the atomic structures of these materials is key to improving their functionality and represents a critical challenge in the field.

In the absence of Bragg diffraction, X-ray and neutron total scattering approaches and, in particular, pair distribution function (PDF) analysis have offered a way to describe, statistically, the average structure of MOFs, from guest molecule ordering to confirmation of continuous random network models for MOF glasses[11,12]. At the other extreme of length scales, moving from the powder scale to that of individual unit cells, high-resolution transmission electron microscopy (TEM) and scanning TEM (STEM) imaging at or near atomic resolution have revealed interface structures[13–15], missing metal centre and missing linker defects[16] and tracked structural changes under the beam with sub-nanometre imaging[17]. However, high-resolution approaches generally require thin samples viewed along high-symmetry orientations and also exhibit limited fields of view, precluding domain characterisation. Figure 1a, b depicts these trade-offs schematically: Powder-scale PDF averages signal over many orientations in a large volume of sample material for high-resolution information in terms of scattering angle. High-resolution STEM (or TEM) maximises spatial resolution at the sample lattice but inherently loses resolution in scattering angle.

A compromise can be struck that offers simultaneously large micron-scale fields of view while retaining nanometre spatial resolution and moderate angular resolution. Figure 1c shows how this can be implemented in a nanobeam scanning electron diffraction (SED) setup, a STEM approach part of the family of four-dimensional STEM (4D-STEM) techniques where the two-dimensional diffraction plane is recorded at each probe position (pixel) in an image[18]. For the SED setup, Bragg spots are well-separated in the diffraction (scattering) plane, recording rich crystallographic detail even for arbitrarily oriented crystals. Amorphous materials contribute to a total scattering pattern which can in turn be transformed for electron PDF (STEM-ePDF) analysis. The SED approach has proven instrumental in the structural characterisation of MOF crystal–glass composites[19–22] and has enabled the visualisation of domain microstructure of defects in UiO-66[23]. In turn, for amorphous MOF composites, the STEM-ePDF modality enables the extraction of PDF signatures comparable to X-ray PDF from nanometre volumes[24]. Crucially, the SED approach enables a low-dose, single-exposure method to minimise beam-induced damage to the sample and simultaneously offers serendipitous identification of small-angle grain boundaries[23] or here, identification of unit cell parameters.

Our focus here is the MOF Fe-BTC (BTC = 1,3,5-benzene-tricarboxylate), known commercially as Basolite® F300. Typically obtained *via* a sol-gel route, Fe-BTC was reported in 2009 for its macroscopically porous aerogel behaviour with a total pore volume of 5.62 cm³ g⁻¹[25]. Since then, Fe-BTC has demonstrated promise as an incredibly diverse heterogeneous catalyst, for example as a Lewis acid catalyst in Claisen-Schmidt reactions, ring opening of epoxides and selective hydrogenations—achieving conversion rates and selectivities of 99% in some cases[26,27]. We recently demonstrated Fe-BTC's promise in the highly sought-after separation of propane and propene, which is currently an incredibly energy-intensive industrial process[28].

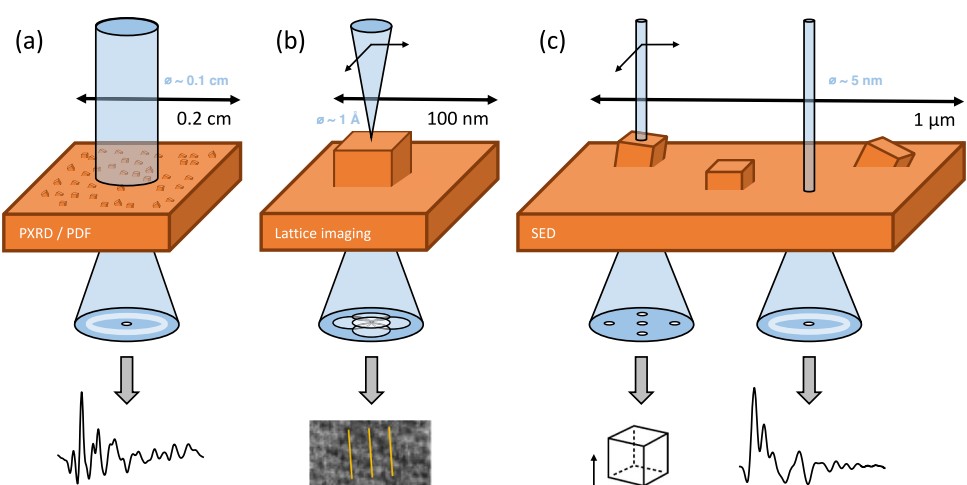

**Fig. 1 Characterisation using scanning electron diffraction. a** X-ray total scattering probes only the average structure. Taking the Fourier transform to obtain the pair distribution function enables us to probe the average local structure. **b** Lattice fringes in HR-STEM probe the local crystallinity, typically over a narrow field of view. **c** Scanning electron diffraction probes the local structure over a wide field of view, providing spatial resolution and good statistical sampling in both real and reciprocal space. Orange cubes represent crystalline domains at a particular orientation embedded within the bulk amorphous matrix.

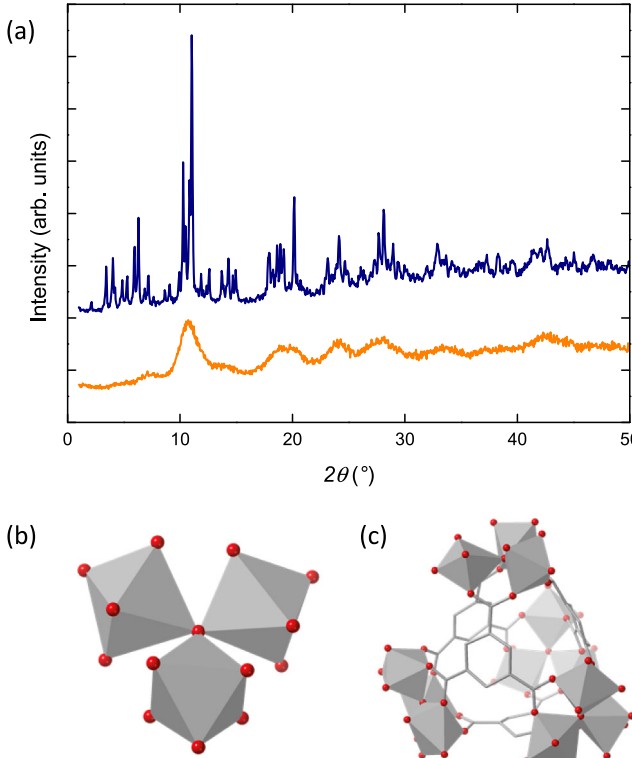

**Fig. 2 Structural chemistry of Fe-BTC. a** Powder X-ray diffraction patterns of MIL-100 and Fe-BTC, adapted from Sapnik et al.[32] (CC-BY 4.0). The **b** $Fe_3O$ trimer unit and **c** tetrahedral assembly of Fe-BTC, adapted from Sapnik et al.[55] (CC-BY 4.0). The $FeO_6$ octahedra are shown as grey polyhedra, with O atoms in red and C depicted as grey wireframe (H atoms omitted for clarity).

Despite its many applications, the structure of Fe-BTC is not fully understood due to its broad diffraction pattern [Fig. 2a]. In 2015, X-ray absorption spectroscopy was used to identify an $Fe_3O$ trimer unit as the fundamental building unit of Fe-BTC, and this was corroborated in 2018 by X-ray pair distribution function measurements [Fig. 2b][29,30]. This same trimer unit is present in MIL-100, a compositionally identical yet crystalline analogue of Fe-BTC[31]. In comparison to Fe-BTC, the structure of MIL-100 is very well understood [Fig. 2a]. In MIL-100, the trimer units assemble *via* the organic linker to form hybrid tetrahedral assemblies [Fig. 2c]. These tetrahedra form the giant-pore network of MIL-100 which has a surface area in excess of 2800 $m^2 g^{-1}$ and a crystallographic unit cell volume in excess of 390,000 $Å^3$ [Supplementary Figs. 1a, b]. Like Fe-BTC, MIL-100 has been investigated for its catalytic abilities, for example in the Friedel-Crafts benzylation of benzene[31]. Despite their similar local structure, the physical properties of Fe-BTC and MIL-100 are very different. Fe-BTC's ability to discriminate between propane and propene is not observed in MIL-100, which adsorbs both gases equally[28]. In catalysis, Fe-BTC out-performs MIL-100 as a Lewis acid catalyst, whereas MIL-100 is favoured for oxidation reactions[26,27]. These functional differences arise from the structural differences between Fe-BTC and MIL-100. Hence, an accurate understanding of Fe-BTC's structure is required to compare to MIL-100.

We recently carried out an extensive structural investigation of Fe-BTC using multiple microscopy (SEM and HR-STEM) and scattering (PXRD, SAXS, WAXS, PDF) techniques[32]. In this earlier study, we found Fe-BTC to possess a nanocomposite structure comprised of an amorphous bulk matrix and a small proportion of nanocrystallites on the order of 200 nm. Small-angle X-ray scattering and HR-STEM suggested these nanocrystallites were unlikely to be domains of MIL-100. Analysis of Fe-BTC's pair distribution function revealed some degree of tetrahedral assembly of the trimer units but not to the same extent as in MIL-100. Using this knowledge we produced several candidate models for the average structure of Fe-BTC using an adapted polymerisation algorithm[33,34]. We found that a model comprised solely of trimer units did not account for the medium-range order in Fe-BTC, while a model built entirely from tetrahedral assemblies overestimated the porosity. Instead, a model containing a mixture of both individual trimer units and assembled tetrahedra captured our experimental results best [Supplementary Figs. 1c, d]. This model was topologically disordered and defective (containing dangling bond defects arising from broken metal–linker bonds).

Despite capturing the average physical behaviour of Fe-BTC very well, this was also a key limitation of our previous work; the model only accounted for the bulk structure of Fe-BTC and not its local variations. Our microscopy data was sensitive to this spatial variation; the X-ray and neutron PDF data were not. In order to fully understand the structure of Fe-BTC, we must independently probe both the structure of the amorphous matrix and the nanocrystalline phases.

Here, we now utilise SED analyses to resolve the spatial variation of both crystalline and amorphous components in the nanocomposite structure of Fe-BTC. The wider field of view enables domain size characterisation beyond what has been possible with lattice imaging. Moreover, we show that classifying diffracting and non-diffracting phases at the nanoscale enables the extraction of ePDF from the amorphous fraction only (rather than an ensemble average obtained in our previous X-ray PDF measurements). SED also reveals additional types of disorder in the crystalline fraction not discernible with other techniques—we report rotational disorder reminiscent of fibre diffraction but now at the nanoscale to reveal ~10° lattice rotations across 10–100 nm length scales. Finally, we use local electron diffraction of single crystals to determine a unit cell for describing the crystalline fraction in Fe-BTC.

## Results

**Mapping the nanocomposite structure.** SED measurements were carried out on identical Fe-BTC and MIL-100 materials as reported previously[32]. The samples were first assessed by examining the particle size and shape. As a 4D-STEM variant, the SED technique enables the reconstruction of virtual annular dark field (ADF) STEM images. With a field of view of ~3 μm in SED, particle morphologies observed in ADF-STEM were broadly similar between MIL-100 and Fe-BTC, consisting of large, likely aggregate, structures ~1 μm or more in size as well as smaller particles and fragments of ~100 nm in size [Supplementary Fig. 2]. These observations were consistent with previous SEM analysis with a wider field of view of 10–100 μm[32].

These morphological observations, however, do not necessarily reflect the crystalline domain size. Here, SED provides access to a large field of view with the specificity proffered by diffraction with a nanoscale electron probe. SED can be used to visualise the crystalline domains through the detection of Bragg scattering across a field of view much larger than observable in lattice imaging by high-resolution TEM or STEM. Moreover, lattice imaging requires crystals to be oriented along specific directions relative to the electron beam to enable the resolution of lattice planes nearly parallel to the incident electron beam. As such, crystals tilted at arbitrary angles with no sets of planes readily resolved (off-axis) have not been observed with prior HR-

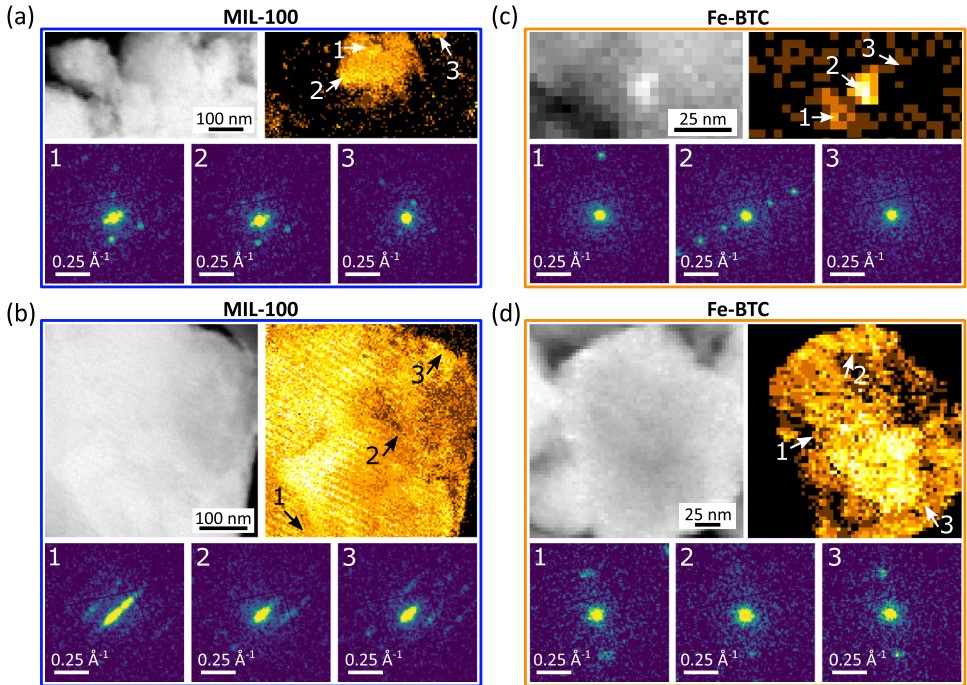

**Fig. 3 Mapping the crystalline domains in MIL-100 and Fe-BTC.** ADF-STEM (grayscale), crystallinity maps (amber), and single-pixel diffraction patterns (Viridis colourmap) from SED datasets acquired on **a**, **b** MIL-100 and **c**, **d** Fe-BTC. The diffraction patterns were extracted from regions marked on the crystallinity maps. Diffraction patterns are displayed as the square root of recorded intensity to enable the simultaneous display of high and low-intensity features.

STEM[32]. Instead, SED is sensitive to any intersection of the sphere inscribed by the elastic scattering condition (incident and scattered wavevectors with equal magnitude, the so-called Ewald sphere) and the reciprocal lattice as modulated by a shape- or thickness-dependent convolutional broadening in reciprocal space. For large unit cell materials with finely spaced reciprocal lattice sites (e.g., those with lattice parameters >7 Å as established for Fe-BTC and MIL-100), the flat Ewald sphere characteristic of high energy electrons will readily satisfy the Bragg condition for arbitrary crystal orientations within scattering angles corresponding to 1 Å$^{-1}$ [32].

In the SED of MOFs, therefore, a reasonable approximation of the location of crystalline and non-crystalline material is achieved by counting detectable Bragg scattering spots in the two-dimensional electron diffraction patterns recorded at every probe position within the field of view. The resulting map of the locations giving rise to Bragg diffraction is termed a crystallinity map[19,35]. Figure 3 presents ADF-STEM, crystallinity maps, and selected single-pixel electron diffraction patterns for MIL-100 and Fe-BTC (see also Supplementary Fig. 3). Figure 3b, d shows domains among the largest observed in each sample. Area-averaged diffraction patterns highlighting indexation of MIL-100 SED data are also presented in Supplementary Fig. 4. The crystalline fraction is revealed as bright intensity in the crystallinity maps. Further inspection of the pixel-wise diffraction enables validation and also identification of domains that are at an approximately constant orientation (a single crystallite) or the identification of multiple distinct crystallites within a field of view. While also granting access to regions of samples significantly thicker than in lattice resolution imaging, SED remains constrained to areas of the sample sufficiently thin for electron transmission (on the order of hundreds of nanometres) and so selected regions of analysis exclude the largest aggregate features where the sample was insufficiently transparent to the electron beam for reliable detection of Bragg diffraction spots.

SED of MIL-100 and Fe-BTC highlighted significant differences in crystalline domain size, extending previous HR-STEM results documenting crystalline domains indexable to MIL-100 and an unknown, likely smaller unit cell crystal structure for Fe-BTC, given the observation of smaller fringe spacings in Fe-BTC over numerous images[32]. SED reveals both large and small crystallites in both MIL-100 and Fe-BTC [Fig. 3], with some crystals as small as a few nanometres and others greater than 100 nm. However, the characteristic observations for the two samples differed in two key ways: (i) on average larger domain sizes and the largest crystalline domains were observed in MIL-100 and (ii) Bragg spots in Fe-BTC were more widely spaced than those in MIL-100, suggesting a smaller unit cell in Fe-BTC.

Figure 4 summarises the domain sizes across more than 20 crystalline domains each in MIL-100 and Fe-BTC as identified by SED crystallinity mapping. These distributions illustrate the larger domain sizes in MIL-100. Figure 3b also highlights the largest domain observed in these datasets for MIL-100, exceeding 800 nm in size. This large crystal also illustrates defined faceting, consistent with a single-crystal character. There is a fringe pattern observed in the ADF-STEM for this particle, attributed to a moiré fringe pattern arising from the overlap of two rigid crystals rotated by a few degrees with identical MIL-100 lattices[36]. This interpretation was supported by inspection of the diffraction pattern integrated over the entire field of view highlighting sets of diffraction spots with a small relative rotation between them [Supplementary Fig. 5]. This moiré fringe indicates a uniform relative rotation between extended single-crystal domains in this particle, further evidence of long-range ordering as expected for MIL-100.

In contrast, Fe-BTC exhibits a strongly skewed distribution of sizes with a majority of observed crystalline domains less than 100 nm in diameter [Fig. 4]. Crystalline domains as small as 10 nm in size were detected as well as larger domains containing broadly consistent single-pixel diffraction patterns across

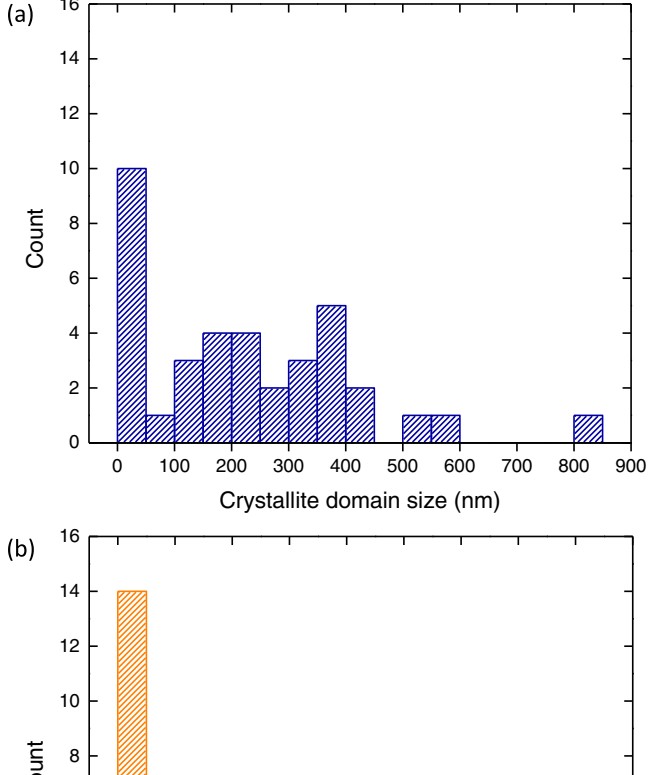

**Fig. 4 Crystallite domain sizes.** Histograms of the crystalline domain size for **a** MIL-100 and **b** Fe-BTC. The crystalline domain represents the average of the longest dimension and the perpendicular width of single-crystal domains observed in crystallinity maps to give a consistent approximation of linear size (see also 'Methods'). Single-crystal domains were defined by inspection of the diffraction patterns.

~100 nm [Fig. 3c, d]. Single-pixel patterns from areas not identified as crystalline show no detectable spots, consistent with the classification of these areas as non-crystalline [pattern 3 in Fig. 3c]. While non-detection of Bragg scattering does not entirely unambiguously determine these areas as non-crystalline, the fact that the detected crystals include finely spaced spots in reciprocal space together with a short wavelength probe makes it highly unlikely 10 nm or larger crystals are undetected. In this approach, there may yet be some false positive labelling as amorphous due to weakly crystalline material or for crystals significantly smaller than 10 nm. Although individual crystalline regions may be on the order of several tens of nanometres in Fe-BTC [Figs. 3 and 4], the positively identified regions showing Bragg diffraction account for a total of 1.2% of the area of the Fe-BTC sample sufficiently thin for SED analysis. This equates to 0.1% by volume if the 1.2% crystalline area is assumed to be spherical volumes viewed in projection.

The Bragg diffraction patterns for the crystalline domains in Fe-BTC exhibited strong, sharp diffraction peaks, consistent with the detection of crystalline domains by HR-STEM lattice imaging

previously[32]. The low symmetry of the patterns, however, indicates these would not be detected in lattice imaging directly due to the imperfect orientation relative to the electron beam for well-resolved lattice plane resolution. The spacing between the spots is also characteristically different to those observed in MIL-100, suggesting a smaller real space characteristic periodicity (smaller lattice parameters) in Fe-BTC. The observed diffraction from Fe-BTC is also consistent with the detection of lattice fringe spacings ≤7.1 Å in Fe-BTC by lattice imaging[32]. In addition, in the Fe-BTC particles of 100 nm or larger size, the diffraction patterns exhibited continuous variation in the orientation of the spots in the plane of the diffraction pattern across the field of view, examined in more detail below (see Disorder in the crystalline domains).

Together, these observations confirm prior analyses of the nanocomposite structure of Fe-BTC with improved sample coverage using material examined at the 10–1000 nm length scale. The observations also now definitively document the domain size of the crystalline fractions. Furthermore, the SED results allow additional evaluation of three scales of (dis)order in Fe-BTC: (i) First, SED enables the isolation of the amorphous and crystalline fractions with electron scattering supporting independent classification and analytical separation of these two fractions. (ii) Next, the nanoscale spatial resolution of SED can be used to reveal rotational disorder across 100 nm in Fe-BTC crystals. (iii) Finally, the Fe-BTC diffraction from single crystals motivates the determination of candidate unit cells for Fe-BTC. These three levels of analysis are now presented in turn.

**Electron PDF analysis.** The electron scattering data recorded in SED approximates total scattering (reasonably complete orientational sampling) to a good degree for non-crystalline materials, enabling concurrent electron pair distribution function (ePDF) analysis. TEM- rather than STEM-based ePDF has seen prior application in amorphous MOFs but without the capacity to isolate mixed crystalline and amorphous phases at the nanoscale, now possible using SED[2]. We have recently demonstrated SED acquisition supports ePDF analysis in MOFs under similar, low-dose electron exposure conditions[24]. SED offers a route to assessing the structure in the amorphous Fe-BTC phases beyond the average amorphous structure observed in X-ray PDF analyses. Instead, the ePDF for the amorphous fraction can be calculated from areas identified as showing no Bragg diffraction.

Figure 5 presents the ePDF analysis for the amorphous regions of Fe-BTC, with comparisons to the bulk X-ray PDF and an average ePDF acquired from MIL-100 crystals. To aid the interpretation of the ePDFs, X-ray PDFs were first calculated for an isolated trimer unit and tetrahedral assembly using a maximum scattering angle $Q_{max}$ equivalent to that achieved in the ePDF measurements [Fig. 5a]. While the scaling between X-ray and electron PDFs is not identical (due to differences in scattering factors), the peak positions remain the same[24]. These calculated PDFs highlight the presence of correlations in the region 1 to 7 Å for the trimer unit and additional correlations between 7 and 12 Å for the tetrahedral assembly.

Figure 5b shows ePDF profiles for Fe-BTC extracted from four separate regions where Bragg scattering was not detected. The individual ePDF profiles, along with their average, consistently show peaks at the same positions. The peak at (i) 1.3 Å arises from C–C and C–O distances, the peak at (ii) 2.0 Å arises from Fe–O distances, the peak at (iii) 3.3 Å from Fe–O and Fe–Fe distances, and the peaks at (iv) 4.7–5.0 Å arise from Fe–O, Fe–C, and C–O distances[32,37]. Subsequent peaks arise from several overlapping contributions.

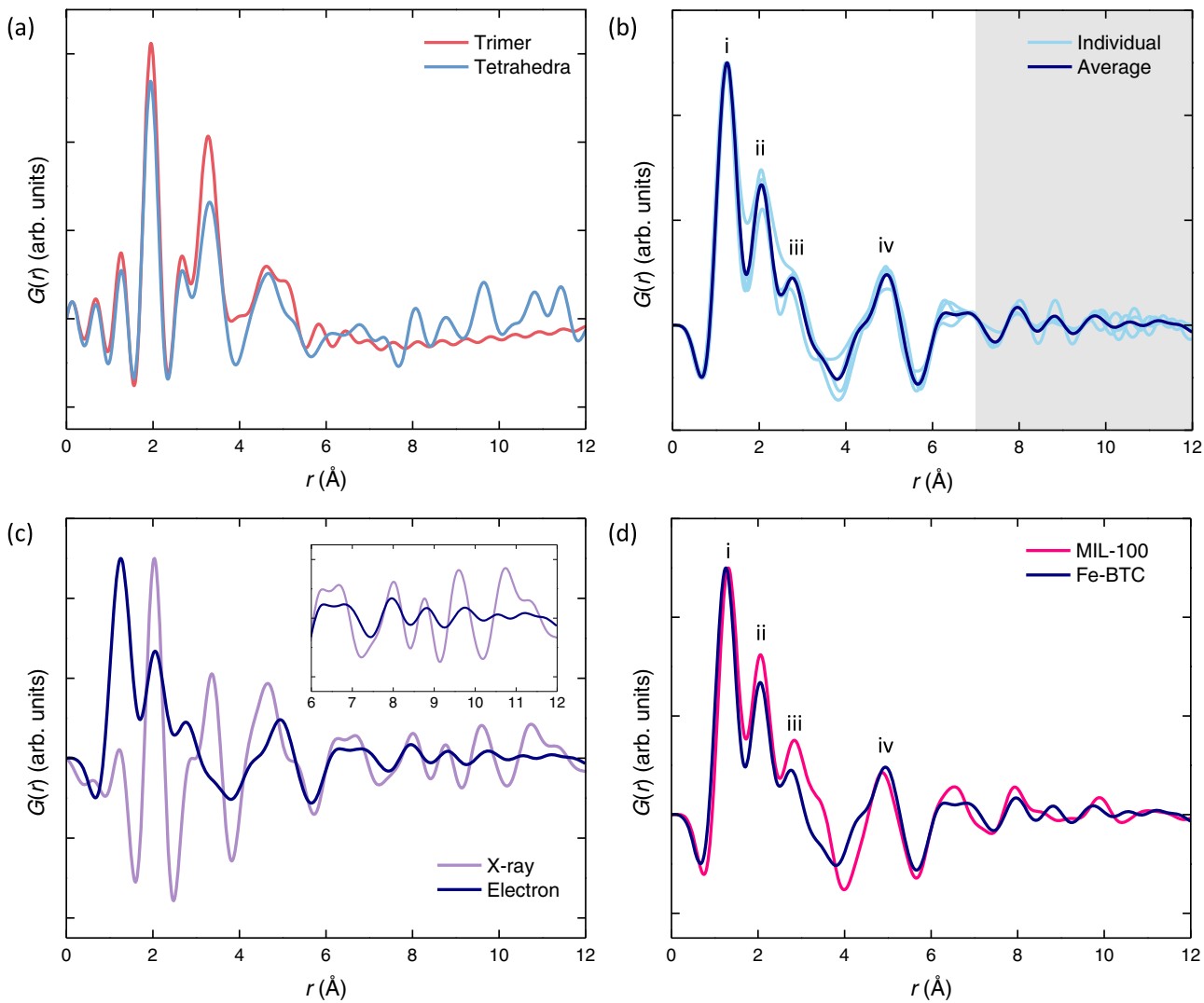

**Fig. 5 Electron PDF Analysis of Fe-BTC. a** PDFs calculated from the trimer unit (red) and tetrahedral assembly (blue) using the Debye scattering equation. **b** Observed ePDF taken as an average of multiple amorphous regions of Fe-BTC (dark blue) alongside ePDF profiles from each of these amorphous regions in the sample (light blue), rescaled to the height of the first peak. The white and grey background regions highlight the regions of the ePDF dominated by trimer and tetrahedral correlations, respectively. **c** Comparison between the X-ray PDF (violet) and the ePDF (dark blue) from (**b**). The X-ray PDF is reprocessed from Sapnik et al.[32] to have the same $Q_{max}$. Profiles were scaled to the same maximum peak height. Inset shows the 6–12 Å region. **d** Average ePDFs obtained from crystalline MIL-100 (pink) and amorphous Fe-BTC (dark blue), rescaled to the height of the first peak.

The ePDFs acquired here are at a much lower maximum scattering angle ($Q_{max} = 11.3 \, Å^{-1}$) than our previous X-ray PDF ($Q_{max} = 23.0 \, Å^{-1}$), necessary to facilitate the simultaneous acquisition of crystalline and amorphous data on a single pixelated detector[32]. As such, our previously collected X-ray total scattering data were reprocessed using a reduced $Q_{max}$ to enable a fairer comparison to the ePDF signal [Fig. 5c]. The peak intensities were not expected to match exactly, as differences in electron and X-ray scattering factors modify the intensity of some pair-wise correlation signals, with particularly low atomic number elements contributing more strongly in ePDF[24] [Supplementary Fig. 6]. Furthermore, the overall intensity of the ePDF profile is thickness-dependent due to significant contributions from multiple and inelastic scattering, and the finite convergence angle of the electron nanobeam also reduces $Q$-resolution, giving rise to a damping envelope with increasing $r$ in the final ePDF[24]. These effects can be minimised by arbitrarily rescaling the ePDFs in intensity to the height of the first peak, as the relative peak intensities are less affected, and peak positions in turn remain

robust with thickness in ePDF[24,38,39]. As a result, we do not think the relative peak intensities are quantifiably comparable, but we find both the ePDF and X-ray PDF for Fe-BTC contain peaks at almost identical positions. Furthermore, the ePDFs for the amorphous fraction of Fe-BTC and the average ePDF acquired for MIL-100 likewise showed agreement in peak positions [Fig. 5d].

The correspondence in peak positions between all three signals (X-ray PDF, ePDF of amorphous Fe-BTC, and ePDF of MIL-100) provides strong direct evidence of the retention of the trimer unit within the amorphous Fe-BTC. Simulated ePDFs likewise show consistency in peak positions with Fe-BTC only for trimer-based structures [Supplementary Fig. 7]. In addition, the strong agreement between bulk X-ray and Fe-BTC ePDF correlations beyond 7 Å indicates the retention of some degree of tetrahedral assemblies within the amorphous Fe-BTC phase, consistent with our previously proposed amorphous model[32]. The slight reduction in peak intensity of some peaks in the ePDF of Fe-BTC compared to those in MIL-100 is consistent with its more

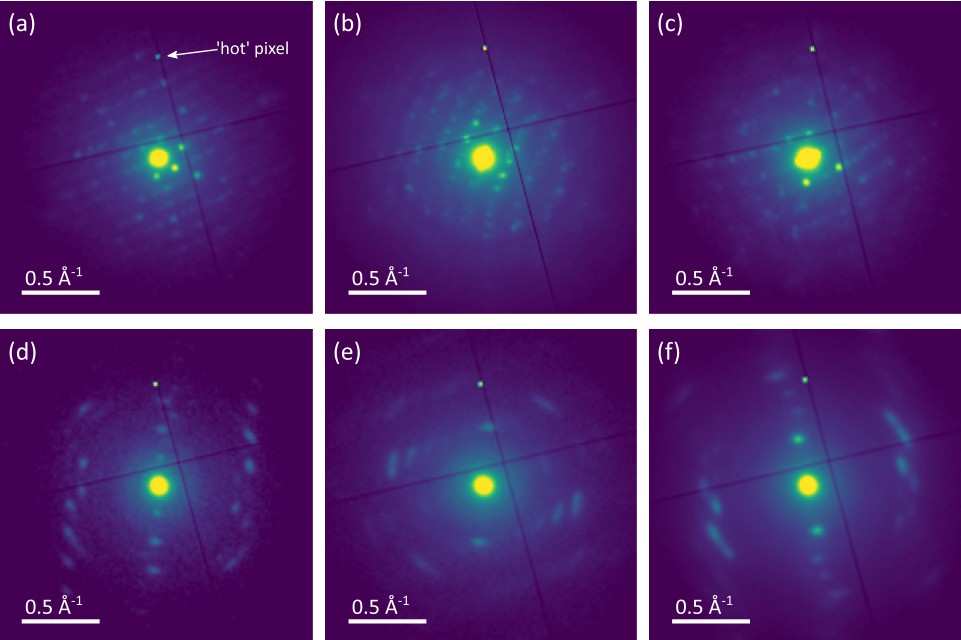

**Fig. 6 Electron diffraction patterns.** Area-averaged diffraction patterns from single crystalline regions in **a–c** MIL-100 and **d–f** Fe-BTC. A hot pixel on the detector appearing on all patterns is marked with a white arrow in (**a**). Diffraction patterns are presented as the square root of recorded intensities to increase visualised dynamic range.

defective nature, particularly for peaks that are dominated by Fe–O correlations (2.0 Å and 3.3 Å)[34].

To obtain a representative ePDF from the corresponding crystalline domains in Fe-BTC, sufficient orientations would need to be sampled within the data. Unfortunately, obtaining a fully representative crystalline ePDF was not possible due to limited crystal orientations observed in Fe-BTC. Nevertheless, by combining all available orientations, we extracted an imperfect crystalline fraction ePDF for Fe-BTC [Supplementary Fig. 7]. The same peak positions were again observed as in the amorphous phase, with the primary difference being stronger peak intensity beyond 8 Å. We avoid further interpretation of this PDF, however, due to the limited orientational averaging of the data.

The similarity between the PDFs presented here indicates a strong correspondence between the local structure of the amorphous and crystalline domains of Fe-BTC, the average local structure of Fe-BTC and the local structure of MIL-100. Subtle differences in peak intensity within ePDFs from different regions in the Fe-BTC sample may indicate nanoscale variations in the degree of tetrahedral assembly both between the crystalline and amorphous phases and within the amorphous phase itself. The ePDFs definitively identify the presence of trimer units and some degree of tetrahedral assembly throughout the structure of Fe-BTC, confirming our previous model for the amorphous component, now with the amorphous Fe-BTC phase isolated spatially using the nanobeam setup. This evidence for the local structure does not, however, explain the striking differences in the diffraction signal obtained between the amorphous and crystalline regions [Fig. 3]. Hence, we subsequently turn our investigation to the regions of Bragg scattering in Fe-BTC.

**Disorder in the crystalline domains**. In addition to single-pixel diffraction patterns that can be extracted from SED datasets [Fig. 3], averaging diffraction patterns over larger areas of the sample improves signal-to-noise in the diffraction data and enables further examination of the degree of crystalline ordering in the diffracting fraction of the sample. Figure 6 presents additional diffraction patterns averaged from crystalline domains

larger than $150 \times 150$ nm in MIL-100 and Fe-BTC. These highlight similar features to those seen in single-pixel diffraction patterns, reproducing the finer spacing of Bragg scattering spots in MIL-100 relative to the coarser spacing typical of Fe-BTC. These observations further corroborate the identification of a likely smaller unit cell for the crystalline material in Fe-BTC.

The spots in MIL-100 were also consistently sharp, reflecting well-ordered crystalline domains. In contrast, Fe-BTC diffraction spots frequently showed arcing about the central beam axis (zero-scattering angle) on averaging across an otherwise single-crystalline domain. This characteristic is similar to features encountered in fibre diffraction in polymeric samples and biological macromolecules with arcs arising from a finite range of misorientations about a common axis[40–42]. To verify whether these arcs observed in Fe-BTC diffraction reflect a rotation of the lattice, Figure 7a, b presents single-pixel patterns from another Fe-BTC particle. While Fig. 7 presents a relatively large particle in the distribution for Fe-BTC [Fig. 4], the arcing is not unique to large particles of Fe-BTC [Supplementary Fig. 8]. The correlated rotation of multiple spots in these single-crystal patterns highlights a predominant in-plane rotational disorder character (whole-pattern rotation) associated with the arcing observed in Fe-BTC [Fig. 7b]. There is accordingly substantial internal disorder and a reduction of the coherent scattering domain size relative to the whole particle size.

There may be further out-of-plane rotation or twisting components, but we focus here on the in-plane component of this rotation as a method for capturing and quantifying the degree of disorder across Fe-BTC samples. Substantial out-of-plane re-orientation (tilting) would, in contrast, lead to the appearance and disappearance of different sharp Bragg scattering vectors from an adjacent section of the reciprocal lattice. The absence of arcing in single-pixel diffraction together with arcs over 10–20° in the average pattern indicates spatially varying rotations of the crystal lattice within the crystalline domain that otherwise shows an approximately constant orientation (consistent set of spots) [Fig. 7c]. These observations further underline the importance of spatially resolved diffraction in the study of

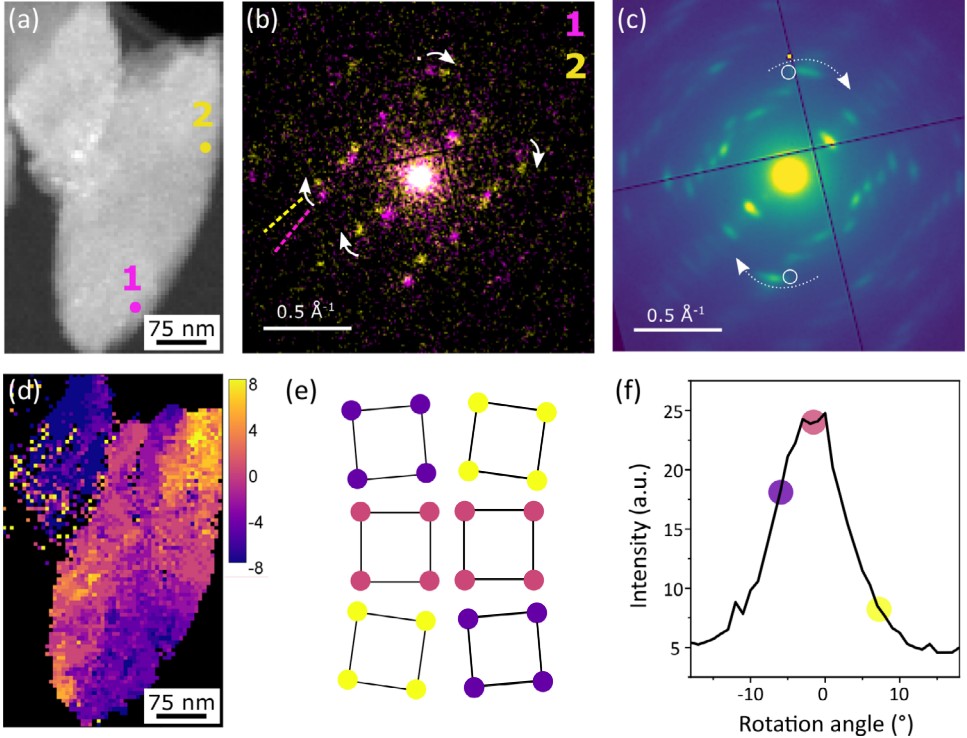

**Fig. 7 Quantifying the rotational disorder in Fe-BTC. a** ADF-STEM and **b** area-averaged diffraction pattern for a crystalline Fe-BTC particle. **c** Rotational orientation map determined from the maximum intensity for a series of Friedel pair virtual apertures, marked as white circles in (**b**), rotated along an arcing reflection at approximately $d^* = 0.6\,Å^{-1}$. **d** Example sets of single-pixel diffraction patterns, marked in (**a**), showing systematic in-plane rotation of the diffraction pattern. **e** Model illustrating the rotations of a simplified representation of the lattice. **f** Intensity profile as a function of rotation angle for selected Friedel pairs' virtual apertures integrated across the entire field of view. Clockwise rotations are taken as positive values. Diffraction intensities are plotted as the square root of recorded intensities to enhance the visualisation of weak and strong diffraction features.

crystalline materials exhibiting mosaicity and other types of disorder at the nanoscale[43].

By tracking the intensity along the arc, we directly measured the angular spread, typically between 10–15° full-width at half-maximum for several particles examined [Fig. 7f and Supplementary Fig. 8] in the integrated Bragg spot (reflection) intensity. The maximum intensity following an arc at each position in the scanned area reflects the relative change in orientation. Plotting this maximum, therefore, provides a visualisation of how the orientations vary spatially across the sample. Figure 7d depicts a continuous evolution of the relative in-plane orientation across the Fe-BTC particle. Each pixel was assessed independently, indicating the systematic, approximately anticlockwise change in rotation angle across the particle.

The regions of the particle with the most positive (clockwise) rotation in the particle appear at the top right and bottom left, with the most negative (anticlockwise) rotations at the bottom right and top left. Intermediate rotation angles appear in regions between these [Fig. 7d, e]. This arrangement shows an approximately diagonal correlation and cross-diagonal anti-correlation in relative rotation angles across the ellipsoidal particle [see also Supplementary Fig. 9], suggesting a degree of orientation cross-polarization emerging from a point near the centre of the crystal. This structure suggests a growth-front linked misorientation resembling a spherulitic growth process common in polymers with attachment and extension rates exceeding re-orientation and ordering timescales. Our observations are also perhaps reminiscent of graphitising carbons, wherein small, ordered domains of layered species grow and coalesce, transforming into a graphitic domain which progressively develops larger structural coherence[44,45]. Such parallels may also provide a

rationale for the apparent log-normal distribution of domain sizes observed in Fig. 4b. A rapid and irreversible solidification mechanism would provide an explanation for weakly ordered crystallisation alongside the formation of amorphous solids to produce an amorphous–crystal nanocomposite form of Fe-BTC. This crystallographic signature is mirrored in the rapid solidification observed during synthesis, motivating the incorporation of mixed trimers and tetrahedra in prior average model structures[32]. The rotational disorder observed within crystals in Fe-BTC, alongside amorphous solids formed in the nanocomposite Fe-BTC structure, show hallmark characteristics of rapid, non-equilibrium Fe-BTC formation mechanisms.

**Unit cell determination.** Given the single-crystal distribution of Bragg diffraction intensity encountered in Fe-BTC particles even as large as 100 nm or higher, we have further sought to derive candidate lattice parameters for these crystals. Due to incomplete orientational sampling as well as additional inelastic and unstructured scattering contributions, fitting to a polycrystalline or azimuthally averaged electron diffraction signal akin to powder diffraction was not feasible despite single-crystal patterns recorded from Fe-BTC crystals [Supplementary Fig. 10]. In particular, the larger Fe-BTC crystals exhibit a sufficient number of spots to derive plausible pairs of basis vectors. Taking things one step further, we recognised the distribution of spots in selected patterns as spanning multiple Laue zones, observed as sets of diffraction vectors grouped in regions of the pattern showing small spot-separations with larger gaps between these zones. These zero, first, and higher-order Laue zones reflect the Ewald sphere cutting through different sections of the three-dimensional reciprocal lattice (additional lattice points above and below those

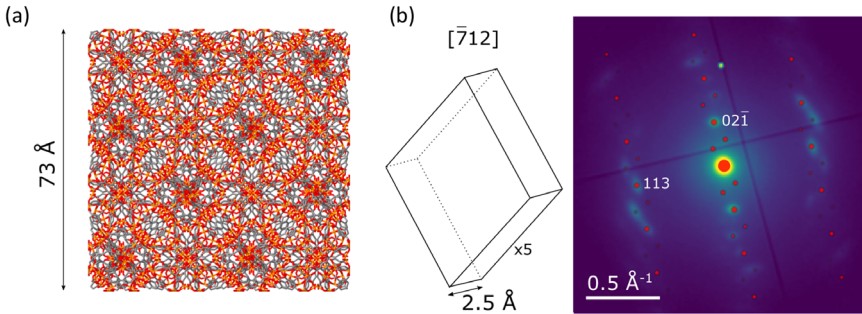

**Fig. 8 Unit cell determination of Fe-BTC. a** Unit cell of MIL-100. Fe (orange), O (red) and C (brown). **b** Candidate unit cell for Fe-BTC (Fe-BTC **1**) determined from the positions of Bragg peaks in the corresponding diffraction pattern (see also Fig. 5f), viewed along the [712] direction. The peak positions for the unit cell are overlaid in red on the experimental diffraction pattern, with experimental intensities plotted as the square root of recorded intensities to simultaneously visualise high and low-intensity features.

**Table 1 Lattice parameters for candidate Fe-BTC unit cells determined from area-averaged SED patterns.**

|  | Fe-BTC 1 | Fe-BTC 2 | MIL-100[31] |
|---|---|---|---|
| $a$ | 2.50 ± 0.088 Å | 2.90 ± 0.022 Å | 73.34 Å |
| $b$ | 7.10 ± 0.060 Å | 7.42 ± 0.25 Å |  |
| $c$ | 7.64 ± 0.14 Å | 10.49 ± 2.1 |  |
| $\alpha$ | 97.8 ± 1.5° | 90° | 90° |
| $\beta$ | 90° |  |  |
| $\gamma$ |  |  |  |

All estimated uncertainties are given as one standard deviation.

contributing to the centre-most group of reflections). As such, these patterns contain both in-plane two-dimensional basis vector information as well as out-of-plane component signatures. A schematic overview of this approach is included in Supplementary Fig. 11. Therefore, three-dimensional lattice parameters can be extracted from the indexation of the experimental data from a single diffraction pattern, following a similar approach to that reported by Shi and Li[46]. In our implementation, we use least squares fitting to first identify a candidate triclinic unit cell. Then, following evaluation of the standardized triclinic cell and identification of any angles in the vicinity of 90°, we have applied monoclinic and orthorhombic constraints on the angles to reduce the number of free parameters [Supplementary Table 1]. Together with inspection of electron diffraction simulations compared against experimental patterns to evaluate the quality of the fit [Supplementary Fig. 12], we report structures with the fewest free parameters that show similar or superior agreement with experimental data to the triclinic parameters.

Figure 8 presents an example of a Fe-BTC diffraction pattern analysed in this way, with a primitive monoclinic unit cell simulated in an overlay on an area-averaged electron diffraction pattern from Fe-BTC. The lattice parameters for this and a further diffraction pattern analysed this way are given in Table 1. An overlay of the experimental data and simulated pattern for the second unit cell is given in Supplementary Fig. 13. The first unit cell was found to be monoclinic, though no space group symmetry was imposed on the recovered unit cell. Due to the disorder identified in the area-averaged Fe-BTC diffraction, the peak positions are smeared into arcs. This rotational disorder across the area included in the average also likely introduces diffraction vectors in addition to those seen at a single orientation of a crystal, contributing a degree of integration across a range of sample orientations. This effect is mimicked in the simulation by increasing the reciprocal lattice rod length, a heuristic adjustment but one that increases the number of reciprocal lattice vectors observed to match the experiment.

As a result of the absence of any imposed symmetry, some simulated diffraction vectors are not observed in the experimental data, likely due to space group symmetry. However, due to the limited data available, we have not sought to attempt symmetry classification. Moreover, any residual electron optical distortions, strain, and imprecision in Bragg spot (disk) centre identification contribute to errors in electron diffraction data. These are not presented as unique solutions but define a minimum unit cell size. Any larger Fe-BTC unit cell or cells will need to be commensurate with one or both of these cells, within the experimental precision.

Both candidate unit cells identified have lattice parameters much smaller than the MIL-100 unit cell, in keeping with previous observations by lattice imaging. Notably, both have a $b$-axis of approximately 7.10–7.42 Å which corresponds to the largest observed $d$-spacing for Fe-BTC by HR-STEM at approximately 7.1 Å (scattering at 12.5° $2\theta$ in Fig. 2a)[32]. Broad scattering below 10° $2\theta$ observed in PXRD of Fe-BTC is consistent with the larger lattice parameter of ~10 Å in the second candidate unit cell [Fig. 2a].

Lattice parameters of approximately 2.5, 7.1, and 7.6 Å bear a resemblance to half- or full-unit cell parameters common to a divalent trimesate crystal (CCDC 171987)[47] and MIL-65 (CCDC 186520)[48], both with a characteristically layered structure. These observations may suggest that the crystalline phase of Fe-BTC could potentially incorporate the trimer units and tetrahedral assemblies within a layered network topology. Electron diffraction from multiple orientations of the same grain could in future extend the unit cell analysis here or lead to structure determination; however, advances would be required to achieve crystal isolation from other nearby crystals or non-crystalline material and to handle the disorder manifest in the arcing Bragg spots toward that end. Unit cell-constrained model-building or structure prediction searches may offer a route to further evaluation of the Fe-BTC crystal structure or structures. Such polymeric structures would also be consistent with possible spherulitic growth suggested by the rotational disorder.

## Conclusions

Electron nanobeam diffraction and pair distribution function analyses provide nano-to-micron scale documentation of the nanocomposite structure of Fe-BTC. We have extracted ePDF analyses to confirm that coordination geometries including Fe trimers and likely tetrahedra are preserved in each of the crystalline and amorphous fractions of the Fe-BTC structure. The Fe-BTC crystalline phases, however, are markedly different from otherwise structurally similar MIL-100, showing extensive rotational disorder in single-crystal domains with significantly smaller unit cells, with estimated candidate unit cell parameters ten times shorter than those of MIL-100. By combining multiple structural

characterisation approaches with spatially resolved analysis from a common set of electron nanobeam measurements, these findings further refine models for Fe-BTC. These advances in MOF characterization in turn encourage wider re-inspection of framework materials for the evaluation of mixed amorphous, crystalline, and multiply disordered structures for the identification of the active, functional component.

## Methods

**Synthesis of MIL-100 and Fe-BTC**. MIL-100 and Fe-BTC were prepared in the same batches as reported previously[32]. For MIL-100, trimesic acid (1.676 g, 7.98 mmol) was dissolved in an aqueous 1 M solution of sodium hydroxide (23.72 g) and iron (II) chloride tetrahydrate (2.260 g, 11.4 mmol) separately dissolved in water (97.2 mL). The linker solution was added dropwise to the metal and left to stir for 24 h at room temperature. For Fe-BTC, trimesic acid (1.1770 g, 5.60 mmol) and iron (III) nitrate nonahydrate (2.5988 g, 6.43 mmol) were each dissolved in 20 mL of methanol. The two solutions were combined and left to stir for 24 h at room temperature. The products were recovered by centrifugation and then washed thoroughly with ethanol before drying overnight at 60 °C. Both products were then purified as described in Sapnik et al.[32], which involved subsequent heating in water, ethanol and aqueous ammonium fluoride. The final powders were activated under a dynamic vacuum at 120 °C overnight.

**Scanning electron diffraction**. Samples were prepared for electron microscopy by drop-casting from a suspension in methanol, where no dissolution was visibly apparent, onto lacey carbon films on copper electron microscopy grids. Samples were rapidly deposited onto the grids, minimising the time spent in suspension, to avoid the possibility of structural rearrangements. For SED data acquisition, a JEOL ARM300CF equipped with a cold field emission gun and aberration correctors in both the probe-forming and image-forming optics at the electron Physical Sciences Imaging Centre (Diamond Light Source, UK), was operated at 200 kV in a custom nanobeam configuration[19,23]. The nanobeam configuration was achieved by turning off the aberration corrector in the probe-forming optics and using a 10 μm condenser aperture to give a convergence semi-angle <1 mrad and a diffraction-limited probe diameter of ~5 nm, as estimated by taking the probe diameter as a disk of radius equal to the distance to the first zero in the Airy probe function. The probe current was measured by a Faraday cup at ~2 pA and the exposure time was set to 1 ms per probe position. The estimated electron fluence in the probe-illuminated sample area was ~5 e$^-$ Å$^{-2}$. Diffraction patterns were acquired using a Merlin-Medipix (Quantum Detectors, UK) camera, a hybrid counting-type direct electron detector. Electron exposure prior to SED data acquisition was minimized by using μs dwell times in STEM for identifying sample areas at low magnification to scan at slower 1 ms exposures.

**Data processing**. SED data were processed using pyxem-0.11.0[49]. First, patterns were rebinned and calibrated prior to analysis. Calibration of the field of view in the scanned areas was carried out using a gold cross-grating standard with a uniform array of gold nanoparticles with a period of 500 nm. The gold cross-grating was also used to determine residual elliptical distortions in the electron diffraction patterns by fitting the polycrystalline ring diffraction pattern from the gold standard. Then, the gold rings were used for diffraction space calibration in $1/d$ units (Å$^{-1}$) following notation common in electron diffraction due to the small scattering angles involved such that $1/d \cong 2\theta/\lambda$ for interplanar spacing $d$, Bragg angle $\theta$ and electron de Broglie wavelength $\lambda$. A $MoO_3$ standard was used to determine the angle of rotation between the diffraction pattern and the scan array. The rotation was corrected to align diffraction patterns to the real space scan image.

ADF-STEM images were formed by integrating across a virtual detector imposed computationally in the four-dimensional SED dataset. Bragg diffraction spots were detected by peak finding using a difference of Gaussians method. The difference of Gaussians method is an image filter implemented in the SciKit Image Python package with the purpose of blurring high-frequency noise and supporting improved peak detection for spot-like or disk-like features. Filter settings were tuned iteratively and assessed by manual inspection to capture disk-like diffraction features in a randomized sub-sample of diffraction patterns followed by application to a full four-dimensional dataset. Crystallinity maps were generated by plotting the number of found peaks at each probe position.

Crystalline domain size was for MIL-100 and Fe-BTC samples were determined from crystallinity maps. Single-crystal domains were identified by inspection of the spot diffraction pattern, with similar positions and arrangements of spots used to classify single-crystal domains. These domains were assessed by manually inspecting diffraction patterns pixel-by-pixel in the four-dimensional dataset from within crystallinity maps. Sets of two or more spots in adjacent pixels at the approximately same orientation were used to identify a candidate single-crystal domain. These were further checked for a consistent number of spots, subject to some spots tilting onto or away from the Bragg condition, to verify the overall spot pattern and orientation of the spot pattern continued across the candidate domain. Once classified, the domain size was measured. The longest dimension of each domain was measured as well as a perpendicular width. The domain size was then taken as the average of these two parameters. The mean aspect ratio (length/width) given by these two parameters was 2.0 ± 0.8 (±1 standard deviation) for MIL-100 and 1.9 ± 0.8 (±1 standard deviation) for Fe-BTC. The average of the length and perpendicular width, therefore, provided a consistent metric for comparison. In order to determine the fractional area accounted for by diffracting domains, the crystallinity maps from all datasets were first thresholded to record the area of the diffracting components. Then, ADF-STEM images were constructed from all areas analysed. These were thresholded to exclude the vacuum and carbon support films as well as to exclude regions that were too thick for diffraction analysis. The threshold for thickness was fixed at a constant across all datasets, determined by inspection of single-pixel diffraction data to identify the ADF-STEM intensity coinciding with an incompletely filled direct beam disk. Regions of the ADF-STEM images that were beyond this threshold showed a weak or lost direct beam, precluding diffraction spot detection. The threshold ADF-STEM images therefore reduced over-counting of these areas not suitable for analysis. The ratio of all diffracting pixel areas and the total area was finally determined as a percentage.

Electron PDF calculations were performed as reported previously[24]. Briefly, area-averaged diffraction patterns were integrated azimuthally using pyFAI within pyXem[50]. Multiple and inelastic scattering contributions were treated by fitting the unstructured scattering profile at a high scattering angle and an additional fourth-order polynomial was included in the fitting procedure after fitting the atomic scattering profile[51,52]. A decaying exponential term of the form $\exp(-bs^2)$ for scattering vector $s$ and adjustable parameter $b$ (typically between 0.6 and 1.0) was used to adjust the scattering profile to zero at the maximum scattering angle prior to Fourier transform calculation[24,39]. In addition, the direct beam (zero-scattering angle) disc was removed and the scattering profile extrapolated to zero from the cut-off imposed by the removal of the direct beam disc. X-ray PDFs for the trimer unit and tetrahedral assembly were calculated using the Debye scattering equation, using a $Q_{max}$ of 11.3 Å$^{-1}$ implemented in DiffPy-CMI[53].

Rotational orientation was quantified by forming virtual dark field images at a selected Bragg scattering vector Friedel pairs[54]. These were then rotated through an arc in 1° steps about the pattern centre with a fixed radius equal to the Bragg scattering vector. The maximum intensity was retrieved over the arc spanning the diffraction spots for each position in the SED dataset to record the in-plane rotation change in the diffraction vector.

Unit cells were determined from diffraction patterns averaged over single-crystal particles. Indexation of the spots to Miller indices $hkl$ was carried out first, by identifying a set of basis vectors explaining the diffraction spot distribution and accounting for gaps between sets of Laue zones crossed by the Ewald sphere. The assigned $hkl$ together with measured $1/d$ values were used to solve the reciprocal lattice metric tensor equation $\mathbf{A}x = 1/d$ for tensor $\mathbf{A}$, $hkl$ values $x$, and the observed inverse $d$-spacings. The equation was solved by least squares minimization implemented in NumPy (Python) and SciPy (Python) to determine primitive lattice parameters in $P1$ symmetry. Unit cells were standardized using Vesta software. Subsequently, monoclinic and orthorhombic constraints were imposed on the angles. Uncertainties were estimated by calculating estimated variances using the curve_fit function in SciPy (Python). Final lattice parameter estimates (Table 1) were evaluated on the basis of using the fewest free parameters to improve uncertainties while verifying a comparable or superior match to experimental data as for the triclinic unit cell by overlaying simulated and experimental patterns.

## Data availability

Data used in this publication are available at the Research Data Leeds Repository at the following link: https://doi.org/10.5518/1269.

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

## Acknowledgements

A.F.S. acknowledges the Engineering and Physical Sciences Research Council (EPSRC) for a PhD studentship award under the industrial CASE scheme, along with Johnson Matthey PLC (JM11106). T.D.B. acknowledges the Royal Society for a University Research Fellowship (URF\R\211013). S.M.C. acknowledges support from the EPSRC (EP/V044907/1). C.S. acknowledges financial support from the China Scholarship Council (CSC) (Grant No. 202006630025). We would like to thank the Diamond Light Source, Rutherford Appleton Laboratory, U.K., for access to ePSIC (EM20198-7). In particular, we thank Dr. Mohsen Danaie for his assistance during our beamtime.

## Author contributions

A.F.S. and S.M.C. designed the project. S.M.C. supervised the project. A.F.S. synthesised the samples and processed the X-ray data. A.F.S., D.N.J. and S.M.C. collected the electron microscopy data. C.S. and S.M.C. performed the analysis of the microscopy data. J.E.M.L. performed the electron PDF analysis. A.F.S., C.S., J.E.M.L., D.N.J., R.B., T.J., P.A.M., T.D.B. and S.M.C. contributed to writing the manuscript. S.M.C., T.J. and T.D.B. acquired funding.

## Competing interests

T.J. works for a company with interests in the commercialisation of MOF materials (Johnson Matthey PLC), and the remaining authors declare no competing interests.
