## [Peer review file · Communications Chemistry]

Mapping nanocrystalline disorder within an amorphous metal–organic frameworkReviewers' comments:

Reviewer #1 (Remarks to the Author):

This paper describes an effective use of scanning electron diffraction methods for analyzing nanostructures of metal-organic frameworks (MOFs). In particular, the electron pair-distribution function analysis is utilized for nanoscale amorphous regions, in addition to the single pattern analysis for nanoscale crystals. This methodology is a powerful tool for revealing complicated nanostructures including both amorphous and crystals, which cannot be analyzed by X-ray total scattering techniques. Although this paper is well written, there are several concerns. My comments are listed below.

1. It is necessary to simulate electron diffraction patterns for MIL-100 and compare them with the experimental patterns (Fig. 6), because the crystal structure of MIL-100 was already solved as shown in Fig. 8. Also, zone axis patterns for Fe-BTC are also needed.
2. The authors mention "Single-pixel patterns from areas not identified as crystalline show no detectable spots, consistent with the classification of these areas as non-crystalline [pattern 3 in Fig. 3c]." on page 7 line 5. However, the absence of diffraction spots does not necessarily lead to the presence of non-crystalline structures. Please reconsider it.
3. In Fig. 5 (c), the difference in $G(r)$ curves between X-ray and electron is too significant. Is this due to differences in atomic scattering factors? If so, could you please show weighting factors for each atomic pair in q -space (reciprocal space)?
4. In my opinion, it is unreliable to derive lattice constants from a single electron diffraction pattern as shown in Fig. 8. Is it possible to obtain multiple patterns from an identical region?
5. It would be better to add more explanations of "MIL-100" in the Introduction for the general readers.
6. In Table 1, the angstrom symbol for MIL-100 is missing.

Reviewer #2 (Remarks to the Author):

This work describes the application of scanning electron diffraction combined to electron pair distribution function analysis to probe the spatially separated atomic structure of an amorphous metal-organic framework, the Fe-BTC. This work is somehow a continuation of previous work from the authors, where they have made significant advances to describe the challenging structure of this disordered material. In this article, the authors go a step further, and give insights now about spatially resolved structural analyses of Fe-BTC. This is a clear and concise article, which shows the potential of e-PDF analyses to probe the atomic structure of disordered MOFs locally. Therefore, my opinion is that this manuscript should be publishable in Nature Communications after minor revision.

Here are my minor concerns about this work:

- The incorporation of e-PDF studies is an interesting point. I have some concerns about the differences in relative peak intensity of the PDF signals, compared to data obtained from X-ray scattering. I understand the intensity of the first peak may be misleading, but there are also significant differences as well in the relative intensity of the following peaks. One concern is how multiple scattering contributions can affect the PDF analysis with electrons. Could the authors expand on this? I think is a point that should be further clarified in the work. Have the authors tried to rescale the ePDFs in intensity to the first peak?
- Sample preparation for the scanning electron diffraction experiments requires the preparation of a suspension with typically an organic solvent (in this case, methanol). I have some concerns about the innocent role of methanol during sample preparation, mostly in terms of structural rearrangements. This would certainly affect reproducibility of the characterization results. While it may be something

unavoidable, I think is important to discuss the reproducibility aspects of such a kind of structural analysis.

Reviewer #3 (Remarks to the Author):

This manuscript reports a scanning electron diffraction (SED) study of Fe-BTC, an important structurally complex MOF that is closely related to the crystalline system MIL-100. In many ways, this study develops from a recent study by many of the same authors (Ref 32) in which a combination of microscopy and diffraction tools were used to interpret the structure of Fe-BTC. The key conclusion of that earlier study was that Fe-BTC is a nanocomposite in which small crystallites were embedded within a disordered amorphous matrix, the structure of which contained key components of the MIL-100 structure (crucially, OFe₃ trimers).

Here, the use of SED generates four dimensional datasets in which 2D diffraction patterns are recorded in a spatially-resolved manner to allow direct interrogation of the diffraction behaviour of different components of the Fe-BTC nanocomposite. For the amorphous component of this nanocomposite, the diffraction signal is inverted to give an electron PDF that is similar to the x-ray PDF measured and interpreted in Ref. 32. For the nanocrystalline component, a unit cell is determined that is different to (smaller than) that of MIL-100.

These are very difficult experiments to carry out, and the nice results presented here will be of great use to the relevant MOF community. I am particularly impressed by the authors' attempts to use SED in a semi-quantitative manner (eg determining crystallinity maps, extracting crystallite size distributions). I expect there is also a strong methodological value in expanding the scientific insight obtained using SED methods. I nevertheless have a handful of questions/comments that the authors might address.

The x-ray diffraction of bulk Fe-BTC samples (Fig 2a) is clearly closely related to that of MIL-100 – intensity maxima being in the same regions of reciprocal space – but with a much reduced effective coherence lengthscale. Of course this is why the corresponding PDFs are similar at low-r. The authors have previously estimated the characteristic coherent lengthscale in Fe-BTC as ~40 nm using the Scherrer equation. Looking at Supplementary Figure 2 in Ref. 32 (from which this value was obtained) I think this value is a vast overestimation that results from the way in which the Pawley method was used there (nb the small peakwidths of component 'ripples' in the fit). A simple estimation from the diffraction peak widths gives a value closer to 40 Å (ie 10x smaller), which is much more consistent with the PDFs in both this paper and Ref 32, and also consistent with the physical extent of amorphous domains within the maps of Fig 3.

The histogram given in Fig 4b documents a different aspect of the Fe-BTC nanocomposite – namely the coherent domain sizes of the separate (small unit cell) nanocrystalline component. The two components are sometimes conflated in the text (eg trying to compare these crystallite sizes to the results of the earlier Scherrer analysis of the bulk amorphous phase). I worry this is misleading.

What seems very important to the story but is never clear to me is the fraction of amorphous and nanocrystalline components in Fe-BTC, which is something one might have hoped SED to determine. My reading of Fig 2a is that the system must be overwhelmingly dominated by the amorphous component. There are no visible peaks in the bulk x-ray diffraction pattern with widths commensurate with the ~100nm domain sizes implied by Fig 4b. Consequently, the small-unit-cell nanocrystalline component must be present in low fractions (ie <1%). Yet the crystallinity map in Fig 3d implies a very significant fraction (what is the meaning of the intensity scale here?) I do not understand how to reconcile the ~40% crystallinity fraction in Fig 3d with the <1% fraction implied by the bulk x-ray diffraction pattern. Is beam damage a problem? Am I misinterpreting Fig 3d? Is this fragment actually

representative of the bulk sample?

Along these lines, perhaps the authors might provide greater detail in SI regarding how the crystallinity maps are actually generated. The methods of Ref 19 suggest that peaks were found using a "difference of Gaussians", but I do not know what this means in this context. The text here implies that the map counts the number of Bragg reflections observed in the diffraction pattern at each pixel position. But what counts as a Bragg reflection? Some more detail would also be valuable regarding the pattern matching used to decide whether neighbouring pixels correspond to the same coherently-scattering crystallite (and hence the domain size distribution function of Fig 4).

If the authors were to generate an 'average' electron diffraction pattern – generated by summing over the diffraction patterns measured at all pixels, and then orientationally averaging this sum – how similar or different is the result to powder x-ray measurement? They should of course be very similar, but if there is any beam damage occurring then the crystalline component may be immediately more obvious. If so, it's possible that determination of its unit cell may actually be easier this way.

The data in Table 1 are very useful for subsequent structural analysis. However, the authors should provide estimated uncertainties obtained from their fitting protocol. Given that the unit-cell angles are not so different to 90° (at least within the reproducibility of the two measurements), perhaps it would be useful to carry out a fit constrained with these to equal 90° - ie assume orthorhombic or higher symmetry. Doing so should improve the certainty on the unit cell lengths.

Reviewers' comments:

Reviewer #1 (Remarks to the Author):

This paper describes an effective use of scanning electron diffraction methods for analyzing nanostructures of metal-organic frameworks (MOFs). In particular, the electron pair-distribution function analysis is utilized for nanoscale amorphous regions, in addition to the single pattern analysis for nanoscale crystals. This methodology is a powerful tool for revealing complicated nanostructures including both amorphous and crystals, which cannot be analyzed by X-ray total scattering techniques. Although this paper is well written, there are several concerns. My comments are listed below.

We thank the reviewer for these positive comments on the effectiveness and power of the presented characterization and on the writing of the paper. We have carefully considered each point raised with responses and associated revisions to the manuscript presented in-line below.

1. It is necessary to simulate electron diffraction patterns for MIL-100 and compare them with the experimental patterns (Fig. 6), because the crystal structure of MIL-100 was already solved as shown in Fig. 8. Also, zone axis patterns for Fe-BTC are also needed.

Our primary purpose in Fig. 6 is to highlight the significantly different features of the patterns recorded from MIL-100 and Fe-BTC without prejudice. We agree with the reviewer that more complete indexation of MIL-100 patterns makes this case even stronger, and so we have added simulation comparisons in the SI, following a reference for the interested reader in the main text:

“Area-averaged diffraction patterns highlighting indexation of MIL-100 SED data are also presented in **Supplementary Fig. 4.**”

Supplementary Figure 4. Area-averaged electron diffraction patterns acquired by SED from MIL-100 crystals shown in (a) Fig. 6b and (b) Fig. 3b. The overlaid simulated diffraction patterns (red) show close correspondence for the principal points. Some additional orientations contribute as a result of area-averaging, giving rise to some additional Bragg spots

that show the same periodicities and are consistent with small rotations or tilts (mosaicity) in this large unit cell structure.

In preparing this revised set of diffraction data we have also corrected a mis-labelled scale bar within what is now Supplementary Fig. 5 in the revised manuscript.

We interpret the reviewer's suggestion of zone axis patterns as meaning low-index or high-symmetry zone axis patterns. In our revised manuscript we have sought to clarify that our approach is rooted in seeking to pursue the retrieval of the maximum amount of information that can be extracted from a single-pass nanobeam diffraction scan in these beam-sensitive materials. The mixed amorphous-crystalline structure of Fe-BTC precludes ready isolation and tilting onto axis of sufficiently large and sufficiently well-ordered single-crystal domains in selected area diffraction mode, and our setup for scanning electron diffraction has a highly limited tilt range. Multi-pass scanning electron diffraction would require a step-change in on-the-fly processing. Such an approach is certainly an exciting prospect and one that we hope to pursue further in future when our data acquisition and data processing hardware and software tools support it.

We have amended the introduction and discussion as follows to clarify our choices:

“Crucially, the SED approach enables a low-dose, single-exposure method to minimise beam-induced damage to the sample and simultaneously offers serendipitous identification of small angle grain boundaries²³ or here, identification of unit cell parameters.”

“Electron diffraction from multiple orientations of the same grain could in future extend the unit cell analysis here or lead to structure determination; however, advances would be required to achieve crystal isolation from other nearby crystals or non-crystalline material and to handle the disorder manifest in the arcing Bragg spots toward that end.”

2. The authors mention “Single-pixel patterns from areas not identified as crystalline show no detectable spots, consistent with the classification of these areas as non-crystalline [pattern 3 in Fig. 3c].” on page 7 line 5. However, the absence of diffraction spots does not necessarily lead to the presence of non-crystalline structures. Please reconsider it.

We agree with the reviewer that absence of evidence should not be construed as evidence of absence. As a result, we chose our wording deliberately to identify consistency with rather than definitive assignment to an amorphous classification.

We have added a sentence to further elaborate this point:

“While non-detection of Bragg scattering does not entirely unambiguously determine these areas as non-crystalline, the fact that the detected crystals include finely spaced spots in reciprocal space together with a short wavelength probe makes it highly unlikely 10 nm or larger crystals are undetected. In this approach, there may yet be some false positive labelling as amorphous due to weakly crystalline material or for crystals significantly smaller than 10 nm.”

3. In Fig. 5 (c), the difference in $G(r)$ curves between X-ray and electron is too significant. Is this due to differences in atomic scattering factors? If so, could you please show weighting factors for each atomic pair in q -space (reciprocal space)?

Differences in intensities in $G(r)$ are a persistent issue for X-ray and electron scattering experiments. However, the positions in r of the pair-wise correlations remains robust. The atomic scattering factors make a contribution to differences, but multiple and inelastic contributions also have a significant affect in electron scattering and cannot be separated or

removed easily in common experimental setups. For completeness, we have added a set of calculated relative scattering factors in Supplementary Figure 6 to highlight the physical differences expected between X-ray and electron scattering which give rise to different relative contributions to scattering by atom pair type as well as in terms of the Q-dependence of these scattering factors.

To further clarify these issues, we have also amended the main text:

“As a result, we do not think the relative peak intensities are quantifiably comparable, but we find both the ePDF and X-ray PDF for Fe-BTC contain peaks at almost identical positions.”

Supplementary Figure 6. Q-dependent relative scattering factors for each pair-pair partial scattering factor, as a fraction of the total scattering factor, for electrons (left) and X-rays (right). The top of each figure is 1 and represents the sum of all pair-pair partial scattering factors. Both X-ray and electron scattering factors are dominated by Fe-based pairs, particularly Fe-Fe scattering. Scattering from C and O based-pairs (such as C-C and O-C in red) contribute substantially less in X-ray scattering than in electron scattering. Additionally, the Q-dependence varies between X-ray and electron scattering.

4. In my opinion, it is unreliable to derive lattice constants from a single electron diffraction pattern as shown in Fig. 8. Is it possible to obtain multiple patterns from an identical region?

We have sought to clarify the methodology used for determining lattice constant estimates from single electron diffraction patterns. We also note that the work by Shi and Li already establishes (Ref. 47) established precedent for our approach.

In addition to the beam sensitivity and hardware limitations on tilting also raised in our response to the reviewer’s first point above, the significant disorder observed in the Fe-BTC crystals presents an intrinsic limitation to obtaining precise information from an identical location from several tilts as the orientations of the crystal vary within the particles.

In conjunction with revisions in response to comments from Reviewer #3 (below), we have now explored further crystal system constraints and include uncertainty estimates on the reported lattice parameters. We have also sought to clarify the assumptions and logic of the approach by including a schematic overview in the supplementary information (Supplementary Figure 11).

Supplementary Figure 11. (Left) Schematic overview of the approach used for the determination of lattice parameters. The reciprocal lattice is shown for an xz plane cut through the three-dimensional reciprocal lattice, with reciprocal lattice points elongated due to the use of thin samples along the beam trajectory. The corresponding diffraction pattern is approximately an xy plane cut through the reciprocal lattice by a nearly planar Ewald sphere (short electron wavelength). (Right) The processing approach is also outlined in sequential steps used to produce and refine lattice parameter estimates.

We have also inserted text in the main text to direct the interested reader to this additional content:

“A schematic overview of this approach is included in **Supplementary Fig. 11.**”

5. It would be better to add more explanations of “MIL-100” in the Introduction for the general readers.

We have expanded our discussion of MIL-100 in the introduction to the following:

“This same trimer unit is present in MIL-100, a compositionally identical yet crystalline analogue of Fe-BTC.³¹ In comparison to Fe-BTC, the structure of MIL-100 is very well understood [Fig. 2a]. In MIL-100, the trimer units assemble *via* the organic linker to form hybrid tetrahedral assemblies [Fig. 2c]. These tetrahedra form the giant-pore network of MIL-100 which has a surface area in excess of 2800 m² g⁻¹ and a crystallographic unit cell volume in excess of 390,000 Å³ [Supplementary Figs. 1a & b]. Like Fe-BTC, MIL-100 has been investigated for its catalytic abilities, for example in the Friedel-Crafts benzylation of benzene.³¹ Despite their similar local structure, the physical properties of Fe-BTC and MIL-100 are very different. Fe-BTC’s ability to discriminate between propane and propene is not observed in MIL-100, which adsorbs both gases equally.²⁸ In catalysis, Fe-BTC outperforms MIL-100 as a Lewis acid catalyst, whereas MIL-100 is favoured for oxidation reactions.^{26,27} These functional differences arise from the structural differences between Fe-BTC and MIL-100. Hence, an accurate understanding of Fe-BTC’s structure is required to compare to MIL-100.”

6. In Table 1, the angstrom symbol for MIL-100 is missing.

We have now added the missing symbol.

Reviewer #2 (Remarks to the Author):

This work describes the application of scanning electron diffraction combined to electron pair distribution function analysis to probe the spatially separated atomic structure of an amorphous metal-organic framework, the Fe-BTC. This work is somehow a continuation of previous work from the authors, where they have made significant advances to describe the challenging structure of this disordered material. In this article, the authors go a step further, and give insights now about spatially resolved structural analyses of Fe-BTC. This is a clear and concise article, which shows the potential of e-PDF analyses to probe the atomic structure of disordered MOFs locally. Therefore, my opinion is that this manuscript should be publishable in Nature Communications after minor revision.

We thank the reviewer for these supportive comments, including on the challenges in characterisation, the clarity of writing, the value of electron beam pair distribution function analyses, and its suitability for publication.

Here are my minor concerns about this work:

- The incorporation of e-PDF studies is an interesting point. I have some concerns about the differences in relative peak intensity of the PDF signals, compared to data obtained from X-ray scattering. I understand the intensity of the first peak may be misleading, but there are also significant differences as well in the relative intensity of the following peaks. One concern is how multiple scattering contributions can affect the PDF analysis with electrons. Could the authors expand on this? I think is a point that should be further clarified in the work. Have the authors tried to rescale the ePDFs in intensity to the first peak?

We have rescaled the ePDFs to the same maximum intensity as the X-ray data in Fig 5c, and to the same intensity in Fig 5d. The electron and X-ray discrepancy unfortunately seems unavoidable given our current instrumentation. Based on previous work (ref 24), we believe this to be a result of four factors. Firstly, different pair-pair scattering factors (we have added Supplementary Fig. 6 to show these) between X-rays and electrons. Secondly, multiple scattering in thicker regions results in an uneven reduction in peak intensities. Thirdly, the finite convergence angle of the beam should result in a dampening envelope in the ePDF, with peaks at higher Q disproportionately damped. Finally, inelastic scattering, which is likely around 50% (or more) of the total scattering, results in an additional signal that modify the calculated ePDF. The overall result is a complex divergence from X-ray intensities that we are not able to disentangle with current instrumentation. To clarify this point (see also response to a related comment by Reviewer 1) we have added the following sentence:

“As a result, we do not think the relative peak intensities are quantifiably comparable, but we find both the ePDF and X-ray PDF for Fe-BTC contain peaks at almost identical positions.”

However, the conditions producing non-linear contributions should be approximately equal between different electron beam scans, and as a result, we believe the comparison to MIL-100 is fair and highlights many similar features.

- Sample preparation for scanning electron diffraction experiments requires the preparation of a suspension with typically an organic solvent (in this case, methanol). I have some concerns about the innocent role of methanol during sample preparation, mostly in terms of

structural rearrangements. This would certainly affect reproducibility of the characterization results. While it may be something unavoidable, I think it is important to discuss the reproducibility aspects of such a kind of structural analysis.

Drop casting from suspension is an effective method to ensure high grid coverage for efficient large area scanning measurements. However, it is important to consider the limitations of this practice as the reviewer suggests. In our samples, MIL-100 was synthesised in water, Fe-BTC in methanol and both washed with ethanol. Water is a particularly poor solvent for drop casting due to its low vapour pressure and high surface tension. The use of water as a drop casting medium would also likely require the use of heating to fully evaporate the solvent before placing under the microscope's vacuum system. This is particularly undesirable as the heating of defective MIL-100 samples in water has been shown to "heal" the defects and increase crystallinity (DOI: 10.1021/acssuschemeng.0c01471). As the reviewer states, organic solvents are more typically used due to their higher volatility which negates the need for heating. Both ethanol and methanol have been used previously to wash MIL-100 and Fe-BTC samples (DOIs: 10.1021/acssuschemeng.0c01471, 10.1039/B910175F and 10.1002/chem.201800694). Given the synthesis solvent for Fe-BTC was methanol, we chose to drop cast from methanolic solutions. In addition, the duration of which the sample is dispersed in methanol is only a few seconds (< 1 min). Furthermore, the boiling point of methanol is over 10°C lower than ethanol which helps to facilitate rapid evaporation. Because of this, we are confident that the structural changes (if any) would be minimal. It is also useful to note that there was no visible dissolution of either sample in methanol. The solvent remained colourless in all cases.

We have added the following in the Methods section:

"Samples were prepared for electron microscopy by drop-casting from a suspension in methanol, where no dissolution was visibly apparent, onto lacey carbon films on copper electron microscopy grids. Samples were rapidly deposited onto the grids, minimising the time spent in suspension, to avoid the possibility of structural rearrangements."

Reviewer #3 (Remarks to the Author):

This manuscript reports a scanning electron diffraction (SED) study of Fe-BTC, an important structurally complex MOF that is closely related to the crystalline system MIL-100. In many ways, this study develops from a recent study by many of the same authors (Ref 32) in which a combination of microscopy and diffraction tools were used to interpret the structure of Fe-BTC. The key conclusion of that earlier study was that Fe-BTC is a nanocomposite in which small crystallites were embedded within a disordered amorphous matrix, the structure of which contained key components of the MIL-100 structure (crucially, OFe₃ trimers).

Here, the use of SED generates four dimensional datasets in which 2D diffraction patterns are recorded in a spatially-resolved manner to allow direct interrogation of the diffraction behaviour of different components of the Fe-BTC nanocomposite. For the amorphous component of this nanocomposite, the diffraction signal is inverted to give an electron PDF that is similar to the x-ray PDF measured and interpreted in Ref. 32. For the nanocrystalline component, a unit cell is determined that is different to (smaller than) that of MIL-100.

These are very difficult experiments to carry out, and the nice results presented here will be of great use to the relevant MOF community. I am particularly impressed by the authors' attempts to use SED in a semi-quantitative manner (eg determining crystallinity maps, extracting crystallite size distributions). I expect there is also a strong methodological value

in expanding the scientific insight obtained using SED methods. I nevertheless have a handful of questions/comments that the authors might address.

We thank the reviewer for the close reading of the manuscript and for the comments noting the difficulty of these experiments, the value of our results to the community, and the important of recovering quantitative results from scanning electron diffraction.

The x-ray diffraction of bulk Fe-BTC samples (Fig 2a) is clearly closely related to that of MIL-100 – intensity maxima being in the same regions of reciprocal space – but with a much reduced effective coherence lengthscale. Of course, this is why the corresponding PDFs are similar at low-r. The authors have previously estimated the characteristic coherent lengthscale in Fe-BTC as ~40 nm using the Scherrer equation. Looking at Supplementary Figure 2 in Ref. 32 (from which this value was obtained) I think this value is a vast overestimation that results from the way in which the Pawley method was used there (nb the small peakwidths of component ‘ripples’ in the fit). A simple estimation from the diffraction peak widths gives a value closer to 40 Å (ie 10x smaller), which is much more consistent with the PDFs in both this paper and Ref 32, and also consistent with the physical extent of amorphous domains within the maps of Fig 3.

The is a very good observation from the reviewer and we agree that the coherence length scale obtained in our previous publication was an overestimation. In this previous work, the Scherrer analysis was used to demonstrate that the scattering pattern from Fe-BTC *could* be interpreted in terms of a broadened MIL-100 diffraction pattern, though it was highlighted that such a refinement alone was insufficient to characterise Fe-BTC as nanocrystalline. As we can see from Fig. 2a in the current work, the diffraction pattern from MIL-100 contains many finely spaced Bragg peaks. When convoluted with the size-dependent broadening term, this led to a particularly unstable refinement. To overcome this, the crystallite size term in our previous work was constrained to 40 nm to give the best visual fit as well as enabling the refinement to converge. As an aside, it is useful to note that the Fe-BTC scattering pattern could likely be refined using any crystalline structure if the broadening term is severe enough.

Given that the crystallite size term was not specifically refined in our previous work, we have removed the following sentence from the current manuscript to avoid any confusion:

“This agrees with our previously reported Scherrer analysis of Fe-BTC’s X-ray diffraction pattern, which hinted at a crystallite domain size on the order of 40 nm, assuming that Fe-BTC could be understood in terms of a nanocrystalline MIL-100 structure.³²”

The histogram given in Fig 4b documents a different aspect of the Fe-BTC nanocomposite – namely the coherent domain sizes of the separate (small unit cell) nanocrystalline component. The two components are sometimes conflated in the text (eg trying to compare these crystallite sizes to the results of the earlier Scherrer analysis of the bulk amorphous phase). I worry this is misleading.

We agree that it is confusing to compare our simple bulk XRD analysis with these more advanced electron diffraction measurements. As per the previous comment from the reviewer, we have deleted the reference to the Scherrer analysis given its limited value.

What seems very important to the story but is never clear to me is the fraction of amorphous and nanocrystalline components in Fe-BTC, which is something one might have hoped SED to determine. My reading of Fig 2a is that the system must be overwhelmingly dominated by the amorphous component. There are no visible peaks in the bulk x-ray diffraction pattern with widths commensurate with the ~100nm domain sizes implied by Fig 4b. Consequently, the small-unit-cell nanocrystalline component must be present in low fractions (ie <1%). Yet

the crystallinity map in Fig 3d implies a very significant fraction (what is the meaning of the intensity scale here?) I do not understand how to reconcile the ~40% crystallinity fraction in Fig 3d with the <1% fraction implied by the bulk x-ray diffraction pattern. Is beam damage a problem? Am I misinterpreting Fig 3d? Is this fragment actually representative of the bulk sample?

We agree that obtaining the ratio of areas or volumes of the nominally amorphous and crystalline components is an important target. In the scanning electron diffraction data, there is an additional concern to avoid mis-classifying areas of the sample that are too thick to show diffraction as entirely non-crystalline regions which complicates a global image analysis. Nevertheless, we have sought to refine an estimate of the nominally amorphous areas within the same areas where the sample is sufficiently thin for unambiguous diffraction identification. We estimate the diffracting regions of Fe-BTC account for 1.2% of the area suitable for analysis, a fractional area that can be conceived as approximately 1 square from a 10×10 array of unit squares (area 100 units).

We have added a comment on these to the results and provided details of our approach in the Methods:

“Although individual crystalline regions may be on the order of several tens of nanometres in Fe-BTC [Fig. 3-4], the positively identified regions showing Bragg diffraction account for a total of 1.2% of the area of the Fe-BTC sample sufficiently thin for SED analysis. This equates to 0.1% by volume if the 1.2% crystalline area is assumed to be spherical volumes viewed in projection.”

“In order to determine the fractional area accounted for by diffracting domains, the crystallinity maps from all datasets were first thresholded to record the area of the diffracting components. Then, ADF-STEM images were constructed from all areas analysed. These were thresholded to exclude the vacuum and carbon support films as well as to exclude regions that were too thick for diffraction analysis. The threshold for thickness was fixed at a constant across all datasets, determined by inspection of single-pixel diffraction data to identify the ADF-STEM intensity coinciding with an incompletely filled direct beam disk. Regions of the ADF-STEM images that were beyond this threshold showed a weak or lost direct beam, precluding diffraction spot detection. The threshold ADF-STEM images therefore reduced over-counting of these areas not suitable for analysis. The ratio of all diffracting pixel areas and the total area was finally determined as a percentage.”

We have also sought to clarify the data in Fig. 3d. There are certainly some large crystals in Fe-BTC, but these are a very small minority. The particle shown in Fig. 7 has an even larger crystal domain all at approximately the same orientation. Importantly, these should be considered within the distribution of crystalline domain sizes, which is strongly skewed towards smaller crystalline domains (Fig. 4). Secondly, these seeming single-crystal domains in fact include many degrees of internal rotational disorder, meaning the overall ‘crystalline domain size’ (all approximately the same orientation) is not a coherent domain of equivalent size. This is perhaps a key example where greater distinction between the coherent scattering size and the apparent single-orientation crystal domains do not reflect the same quantity.

We have amended the text to clarify these points:

“**Figure 3** presents ADF-STEM, crystallinity maps, and selected single-pixel electron diffraction patterns for MIL-100 and Fe-BTC (see also **Supplementary Fig. 3**). **Figures 3b & d** show domains among the largest observed in each sample.”

“To verify whether these arcs observed in Fe-BTC diffraction reflect a rotation of the lattice, **Figures 7a & b** present single-pixel patterns from another Fe-BTC particle. **While Figure 7** presents a relatively large particle in the distribution for Fe-BTC [**Figure 4**], the arcing is not unique to large particles of Fe-BTC [**Supplementary Fig. 8**]. The correlated rotation of multiple spots in these single-crystal patterns highlights a predominant in-plane rotational disorder character (whole-pattern rotation) associated with the arcing observed in Fe-BTC [**Fig. 7b**]. **There is accordingly substantial internal disorder and a reduction of the coherent scattering domain size relative to the whole particle size.**”

Along these lines, perhaps the authors might provide greater detail in SI regarding how the crystallinity maps are actually generated. The methods of Ref 19 suggest that peaks were found using a “difference of Gaussians”, but I do not know what this means in this context. The text here implies that the map counts the number of Bragg reflections observed in the diffraction pattern at each pixel position. But what counts as a Bragg reflection? Some more detail would also be valuable regarding the pattern matching used to decide whether neighbouring pixels correspond to the same coherently-scattering crystallite (and hence the domain size distribution function of Flg 4).

We have amended the text to provide more details on the methods:

“Bragg diffraction spots were detected by peak finding using a difference of Gaussians method. The difference of Gaussians method is an image filter implemented in the SciKit Image Python package with the purpose of blurring high frequency noise and supporting improved peak detection for spot-like or disk-like features. Filter settings were tuned iteratively and assessed by manual inspection to capture disk-like diffraction features in a randomized sub-sample of diffraction patterns followed by application to a full four-dimensional data-set. Crystallinity maps were generated by plotting the number of found peaks at each probe position. Single-crystal domains were identified by inspection of the spot diffraction pattern, with similar positions and arrangements of spots used to classify single-crystal domains. These domains were assessed by manually inspecting diffraction patterns pixel-by-pixel in the four-dimensional dataset from within crystallinity maps. Sets of two or more spots in adjacent pixels at the approximately same orientation were used to identify a candidate single-crystal domain. These were further checked for a consistent number of spots, subject to some spots tilting onto or away from the Bragg condition, to verify the overall spot pattern and orientation of the spot pattern continued across the candidate domain. Once classified, the domain size was measured. The longest dimension of each domain was measured as well as a perpendicular width.”

If the authors were to generate an ‘average’ electron diffraction pattern – generated by summing over the diffraction patterns measured at all pixels, and then orientationally averaging this sum – how similar or different is the result to powder x-ray measurement? They should of course be very similar, but if there is any beam damage occurring then the crystalline component may be immediately more obvious. If so, it’s possible that determination of its unit cell may actually be easier this way.

We agree that given sufficient orientational averaging (i.e. a polycrystalline ring pattern), powder X-ray diffraction measurements and azimuthally averaged one-dimensional electron diffraction profiles should offer good comparisons. In this case, we are collecting electron diffraction data over a very different angular range (range of momentum transfer).

Nevertheless, we have combined all of the Fe-BTC diffraction patterns as for our ePDF analysis (Supplementary Figure 7). We now include the azimuthally averaged one-dimensional profile for this combination of Fe-BTC diffraction patterns as Supplementary Figure 10. Both in the two-dimensional sum and as a one-dimensional profile, the diffraction data shows substantially incomplete sampling. We have also included the same profile with

unstructured scattering contributions subtracted (the input to ePDF calculations). We note the comparison with powder X-ray diffraction data is unfortunately not straightforward due to the differences in momentum transfer examined as well as the significant contributions from unstructured scattering in the electron beam data (which is much more substantial than in X-ray data) and significant inelastic scattering contributions to the electron scattering profile.

We have amended the text to note this alternative approach in the context of unit cell determination:

“Due to incomplete orientational sampling as well as additional inelastic and unstructured scattering contributions, fitting to a polycrystalline or azimuthally averaged electron diffraction signal akin to powder diffraction was not feasible despite single crystal patterns recorded from Fe-BTC crystals [Supplementary Fig. 10]. In particular, the larger Fe-BTC crystals...”

Supplementary Figure 10. (a) Summed diffraction patterns of the crystalline Fe-BTC fraction. This pattern, effectively an average pattern, exhibits strong ‘spottiness’ indicative of poor sampling. (b) The corresponding azimuthal profile, acquired by integration around the direct beam. The cross present due to gaps in the detector was removed prior to azimuthal integration. (c) The profile with unstructured scattering subtracted, the $\phi(Q)$ input used for ePDF calculation. The upper horizontal axes show the corresponding 2θ range (Cu K α 1 = 1.5406 Å).

The data in Table 1 are very useful for subsequent structural analysis. However, the authors should provide estimated uncertainties obtained from their fitting protocol. Given that the unit-cell angles are not so different to 90° (at least within the reproducibility of the two measurements), perhaps it would be useful to carry out a fit constrained with these to equal 90° - ie assume orthorhombic or higher symmetry. Doing so should improve the certainty on the unit cell lengths.

We thank the reviewer for this incredibly useful suggestion. We have extended our approach to take the standardized triclinic lattice parameters as a starting point for constrained lattice parameter estimation for monoclinic or orthorhombic crystal systems. We have also now extracted variances during the least squares minimisation and report estimated standard deviations for each parameter. We now report the first candidate unit cell (Fe-BTC **1**) as a monoclinic unit cell and the second candidate unit cell (Fe-BTC **2**) as an orthorhombic unit

cell. For completeness we include all triclinic, monoclinic, and orthorhombic cells in Supplementary Table 1 in the revised manuscript.

To enact these changes in the manuscript, we have amended Table 1:

Table 1 Lattice parameters for candidate Fe-BTC unit cells determined from area-averaged SED patterns. All estimated uncertainties are given as one standard deviation.

	Fe-BTC 1	Fe-BTC 2	MIL-100 ³¹
a	2.50 ± 0.088 Å	2.90 ± 0.022 Å	
b	7.10 ± 0.060 Å	7.42 ± 0.25 Å	73.34 Å
c	7.64 ± 0.14 Å	10.49 ± 2.1	
α	97.8 ± 1.5°		
β		90°	90°
γ	90°		

We have also amended the text on lattice parameter estimation:

“In our implementation, we use least squares fitting to first identify a candidate triclinic unit cell. Then, following evaluation of the standardized triclinic cell and identification of any angles in the vicinity of 90°, we have applied monoclinic and orthorhombic constraints on the angles to reduce the number of free parameters [Supplementary Table 1]. Together with inspection of electron diffraction simulations compared against experimental patterns to evaluate the quality of the fit [Supplementary Fig. 12], we report structures with the fewest free parameters that show similar or superior agreement with experimental data to the triclinic parameters.”

We have made minor wording changes in the following text as well to reflect the revised results. And we have added additional Methods text:

“The equation was solved by least squares minimization implemented in NumPy (Python) and SciPy (Python) to determine primitive lattice parameters in *P1* symmetry. Unit cells were standardized using Vesta software. Subsequently, monoclinic and orthorhombic constraints were imposed on the angles. Uncertainties were estimated by calculating estimated variances using the curve_fit function in SciPy (Python). Final lattice parameter estimates (Table 1) were evaluated on the basis of using the fewest free parameters to improve uncertainties while verifying a comparable or superior match to experimental data as for the triclinic unit cell by overlaying simulated and experimental patterns.”

We have updated Figure 8 in the revised manuscript to show the monoclinic unit cell for Fe-BTC 1:

Fig 1 Unit cell determination of Fe-BTC (a) Unit cell of MIL-100. Fe (orange), O (red) and C (brown) (b) Candidate unit cell for Fe-BTC (Fe-BTC 1) determined from the positions of Bragg peaks in the corresponding diffraction pattern (see also Fig. 5f), viewed along the $[\bar{7}12]$ direction. The peak positions for the unit cell are overlaid in red on the experimental diffraction pattern, with experimental intensities plotted as the square root of recorded intensities to simultaneously visualise high and low-intensity features.

We have also added corresponding Supplementary Fig. 12-13 to show comparisons between simulated and experimental patterns to evaluate the determined candidate unit cells.

Supplementary Figure 12. Overlays showing experimental diffraction data and kinematical simulations of electron diffraction for (a) triclinic, (b) monoclinic, and (c) orthorhombic crystal systems after least squares minimisation to identify lattice parameters for Fe-BTC 1. The constrained fitting reduces apparent uncertainties (Supplementary Table 1), but the orthorhombic constraints for Fe-BTC 1 appear to reduce the quality of the fit with experimental data. White arrows highlight points in the orthorhombic-constrained unit cell that show inferior agreement with experimental data.

Supplementary Figure 13. (a) Second candidate unit cell for Fe-BTC (Fe-BTC **2**) determined from the positions of Bragg peaks in (b) the corresponding diffraction pattern (see also **Fig. 7b**), viewed along the $[\bar{1}1\bar{2}]$ direction. In (a)-(b) the lattice parameters were determined with no constraints in the triclinic crystal systems (Fe-BTC **2a**). (c) Candidate unit cell for Fe-BTC **2** with orthorhombic crystal system constraints applied with (d) an overlay of simulated and experimental diffraction patterns showing a comparable match with fewer free parameters. The peak positions for the unit cell are overlaid in red on the experimental diffraction pattern, with experimental intensities plotted as the square root of recorded intensities to simultaneously visualise high and low-intensity features.

REVIEWERS' COMMENTS:

Reviewer #1 (Remarks to the Author):

The authors' response and revisions are satisfactory. I believe the paper is now publishable in Communications Chemistry.

Reviewer #2 (Remarks to the Author):

I would like to thank the authors for their detailed response to the reviewers' points. The revised version has satisfied all of my concerns regarding this manuscript.

Reviewer #3 (Remarks to the Author):

The authors have done an excellent job of addressing all points raised by the referees, and I am happy to recommend publication of the revised manuscript.